# Mechanical Characterization of Flax and Hemp Fibers Cultivated in Romania

**DOI:** 10.3390/ma17194871

**Published:** 2024-10-03

**Authors:** Constantin Stochioiu, Miruna Ciolcă, Anca-Loredana Deca

**Affiliations:** Strength of Materials Department, Faculty of Industrial Engineering and Robotics, National University of Science and Technology POLITEHNICA Bucharest, 060042 Bucharest, Romania; anca_loredana.deca@stud.fiir.upb.ro

**Keywords:** hemp fiber, flax fiber, mechanical testing, Young’s modulus, Weibull modeling

## Abstract

This study examines the mechanical properties, specifically strength and stiffness, of technical hemp and flax fibers grown in Romania. Tensile testing was employed to determine stress–strain curves and the Young’s modulus and to assess the failure strength of both fiber types. Although samples of various lengths were tested, no significant length-dependent variations were observed. However, a strong dependence on fiber diameter was noted, with the smallest diameters approaching the documented strength of elementary fibers. Due to the considerable variability in the experimental results pertaining to the characteristics of the reinforced fibers, a statistical analysis using a two-parameter Weibull distribution was employed. The analysis revealed three distinct stress–strain curve profiles, i.e., linear, bi-linear, and tri-linear patterns, with the average ultimate stress ranging from 412 to 566 MPa for hemp and 502 to 598 MPa for flax.

## 1. Introduction

In an effort to minimize environmental impact, the composite material industry is increasingly focusing on replacing synthetic materials, either partially or entirely, with renewable and sustainable alternatives [1,2]. Recently, vegetable fibers have emerged as effective reinforcements [3], with numerous studies demonstrating the potential of composites made from these fibers to achieve significant mechanical properties [4]. This success has confirmed the potential of plant-based composite materials and has driven the development of components made from these sustainable sources, leading to an estimated market size of EUR 5.2 billion compared to the EUR 47.4 billion market size of the general composite industry in 2023 [5,6].

The trend of adopting vegetable fiber reinforcements is prominent in regions where the source plants can be cultivated, providing a local and sustainable source of raw materials. In Europe, flax and hemp stand out as notable examples [7,8].

Flax, *Linum usitatissimum*, is a flowering plant from the family of Linaceae. It is a highly versatile plant that requires a low amount of input [9]. It can be cultivated in temperate climates and matures in approximately 100 days [9]. The flax plant can grow up to 1.2 m high, with few ramifications. While the uses of the plant are remarkably diverse, ranging from food source or pharmaceuticals to paint oils [10], its main use is in the textile industry, where its fibers are used to make linen, a high-quality fabric [10].

Hemp belongs to the *Cannabis sativa* species and originates from East Asia. It is a fast-growing, high-yield crop with minimal cultivation requirements [11]. Through selective breeding, it has developed a wide range of traits, enabling its diverse use across various industries, similar to flax. A distinctive characteristic of hemp is its production of tetrahydrocannabinol (THC), which varies in concentration depending on the cultivar [12]. Due to this trait, hemp cultivation is subject to different regulations, with some countries banning it and others enforcing strict laws [13,14]. In Romania, hemp cultivators must register and conduct periodic analyses of THC content in their crops [15].

Between 2015 and 2022, the global average annual production of hemp was 189.5 thousand tons and that of flax was 885.5 thousand tons [16].

Historically, flax and hemp were significant crops in Romania, with annual production ranging from 20 to 40 thousand tons for both plants during the communist era, due to the high demand from the textile industry, a key driver of the economy at that time [16]. This period also saw the development of local cultivars, such as Rolin, which were well adapted to local conditions [17].

However, production dramatically declined in the late 1980s, with flax reaching just 106 tons and hemp to 2600 tons in the 2015–2022 period [16]. The drop can be attributed to several factors, the primary one being a reduced demand from the textile industry, which globally shifted toward synthetic fibers and cotton [14].

In response to this shift, a global trend has emerged to repurpose these raw materials for other industries, such as construction [18] and composite materials [19]. This trend does not appear to be significant for Romania, as indicated by the reduced production volumes.

Flax and hemp are sought after in the composite industry mainly due to their high mechanical properties and low density, with several studies having revealed comparable values in terms of specific strength and especially stiffness to glass fibers [20,21]. These qualities come from the high content of cellulose present in the structure of the fibers. However, their mechanical properties have a high variability, mainly attributed to the cultivar, cultivation regime (region, weather, soil, etc.), extraction technique, and fiber length [22,23].

Apart these qualities, flax and hemp fibers have several other advantages, such as those mentioned below:-They are derived from renewable resources and have a low environmental impact when compared to synthetic reinforcements [9];-The technology for extracting the fibers is well established, with only minor adjustments required to cater to industry-specific needs, such as adding equipment for producing rovings, mats, or resin pre-impregnated rolls in the fabrication process;-They are biodegradable, allowing the creation of a fully biodegradable composite if an appropriate matrix is used [24];-They are low cost and have low energy requirements for production [9,25];-They possess better fiber/matrix bonding properties when compared to other vegetable fibers [26];-They pose low health risks for workers involved in production [27].

In Western and Northern Europe, the natural fiber industry, extending beyond textiles, is robustly supported by environmental policies and local agricultural protections. In contrast, the industry in Romania has diminished in scale [28], despite possessing similar potential and favorable climatic conditions in regions such as the Olt Valley, Central Transylvania, and the northern regions [28].

Given the current market conditions, which could potentially support competitive pricing, there is a clear justification for examining the mechanical properties of Romania’s flax and hemp fibers to determine their effectiveness in reinforcing structural composite materials. Data on the mechanical properties of technical fibers are rather scarce, making it particularly important to focus on these fibers. Mechanical testing of technical fibers is crucial, as these fibers are typically used in the bulk form in composites, where their collective behavior significantly impacts the performance of the final material. Unlike elementary fibers, which are often studied in isolation [29,30], technical fibers more accurately represent real-world applications, where they are bundled and used together. Understanding their mechanical properties can help optimize the design and manufacturing processes of composite materials, ensuring better performance, reliability, and cost-efficiency in structural applications.

The focus of this study is on analyzing hemp and flax technical fibers, specifically examining their stress–strain curves, strength, and stiffness through two-parameter Weibull statistical modeling. Additionally, the study explores the effects of fiber length and cross-sectional area.

The study is important for a region with considerable potential in terms of eco-friendly reinforcements, as it could open new avenues for development in both the industry and research sectors. It may also help align the Romanian hemp and flax industry with European and global movements toward utilizing high-quality, eco-friendly composite materials. Additionally, considering the significant variability in the results reported in the literature, contributing additional data on the mechanical properties of these fibers to the existing database is particularly valuable.

### 1.1. Fiber Structure

Both flax and hemp fibers are extracted from the stem of the plant in several steps, through biological (retting or chemical treatments) and mechanical means (decortication, scutching, etc.) [31,32].

They share a generally similar structure, with their elementary fibers exhibiting a complex microstructure and geometry [33,34]. The fibers are situated on the outer wall of the plant stem (Figure 1a), where they are organized into bundles. These bundles are held together by amorphous polymers, primarily pectin (Figure 1b).

The elementary fiber structure is layered and hierarchical (Figure 1c) and is organized into two cell walls, the outer one, called the primary cell wall, having a protective role, and the inner one, the secondary cell wall, which is thicker than the primary cell wall, having a structural role. In the center, a void is located, called a lumen. The secondary cell wall is mainly composed of crystalline cellulose, in the form of microfibrils and amorphous hemicellulose, acting as a matrix. The microfibrils are spirally oriented along the fiber axis, at a microfibrillar angle, α. The shape of the elementary fiber is polygonal in cross-section, while the fiber bundle can be even more complex, depending on the size, number of elementary fibers, and organization [36,37]. Due to this complexity, the cross-section is often approximated to be rectangular [35], elliptical [38], or circular, with circular being the most common approximation [22].

From a structural point of view, the main differences between the two fibers are the percentage of their constituents and the angle at which the cellulose microfibrils are wound along the fiber length [20].

In mechanical testing, the complex shape of the fibers is known to significantly impact the results, leading to considerable variation in the calculated properties. Several other documented factors influence these variations and include the following:The helical arrangement of microfibrils generates a complex stress state under load.The extraction process may alter the structure, thus reducing its mechanical properties.Kink bands, which are local defects, reduce the structural integrity of the fibers [39].Small-diameter fiber bundles tend to twist themselves when extracted from a plant, leading to difficulties in measuring their cross-section and introducing additional stresses during testing. An example of such fiber bundles is given in Figure 2 and was obtained by the authors through optical microscopy.

Due to these factors, a significant number of approximations are required for utilizable results on the mechanical response of vegetable fibers. However, the approximation approach, even if conventional, is the most widely accepted approach in the study of this type of fibers [40], since it is the most straightforward approach and it generates usable information about vegetable fibers for the reinforcement of composite materials.

### 1.2. Related Studies

Studies on natural fibers are presented in the literature, covering various varieties, regions, and methods of cultivation or extraction. However, only one study has been identified on elementary or technical fibers originating from crops cultivated on the Romanian territory [41]. Moreover, the studies are predominantly focused on elementary fibers, with scarce information available on technical fibers or fiber bundles. In light of this, a brief review of data from the literature on the strength and stiffness of hemp and flax fibers is provided in Table 1 and Table 2.

The review excludes data on fibers that have undergone chemical treatments post-extraction, as such treatments fall outside the scope of the current study.

A significant variability is evident both within individual studies and across different studies. However, a clear relationship between fiber diameter and tensile strength is observed in flax fibers, with elementary fibers reaching strength values of up to 1500 MPa. Although the data for hemp fibers are quite limited, they suggest a similar trend.

## 2. Materials and Methods

Mechanical testing was conducted on fibers from two different sources, each provided as ballots: one of hemp, supplied by HempFlax, Sebeș, Romania, and one of flax, provided by Faltin, Fălticeni, Romania, (Figure 3a,b). The hemp was cultivated in the region of Sebeș, Alba County, in 2023, while the flax was cultivated in Suceava county, in 2022. Both fibers were mechanically extracted as short fibers, primarily intended for use in thermal insulation.

### 2.1. Mechanical Testing

Testing is conducted in accordance with ASTM C1557 [50], in which fibers are individually subjected to tensile testing up to failure. Due to the high variability in mechanical properties, which is already documented for all types of fibers, i.e., synthetic [51] and natural fibers [52], a statistical approach was required. Furthermore, fiber length has been documented from several sources as being a factor of influence [29,51]. Consequently, four fiber lengths were used, namely, 10 mm, 15 mm, 20 mm, and 25 mm, with 25 samples per batch, leading to a total of 200 tested samples (Table 3).

Before sample preparation, the fibers were cleansed using a soap and water solution to minimize impurities and the amount of potentially degrading bacteria. Even though such a procedure is absent in the C1557 norm, it is a frequent practice in the retting process of bast fibers [53].

The samples were prepared by extracting the fibers from the ballots, with a tweezer. The aim was to extract bundles with few elementary fibers. To minimize the mechanical stress during extraction, the fibers were soaked in water for three to seven days to temporarily soften the pectin bonds [53], thereby making it easier to extract fiber bundles with smaller cross-sections.

The extracted fibers were glued to a cardboard support, where they were left to dry for two to four days before measuring. The tabs are presented in Figure 4a and are laser cut based on the previously described gauge lengths. The set of small circular holes was used to allow the centering of the fiber in the paper support, while the bigger, outer holes allowed for the centering of the paper support in the testing machine’s grips.

After sample preparation, the cross-sections of the fibers were individually determined at five points along the calibrated length (Figure 4b), with the help of an Insize ISM M1000 optical microscope (Atlanta, GA, USA) equipped with an INSIZE ISM D500 digital camera (Insize, Atlanta, GA, USA). For the present study, the cross-section was considered circular, ignoring the lumen.

Testing was conducted using an in-house miniature tensile testing machine, Figure 5, equipped with an HBM 100 N load cell (HBM, Darmstadt, Germany) and an HBM WA10 displacement transducer (HBM, Darmstadt, Germany). The tests were performed under displacement control at a speed of 1.5 mm/min. Data acquisition was carried out using an HBM QuantumX data acquisition system (HBM, Darmstadt, Germany) at a frequency of 50 Hz.

The conventional stress, σ, was calculated as the ratio between the recorded force, F, and the minimum cross-section of the fiber, A, as in Equation (1), while the strain, ε, was calculated as the ratio between the displacement, ΔL, and the initial length, $L_{0},$ as in Equation (2).
(1)$$\sigma =\frac{F}{A}$$
(2)$$\varepsilon =\frac{\Delta L}{L_{0}}$$

Strain calculation was performed by applying two additional considerations. First, the fiber cross-section was assumed to be constant along its length, allowing the use of Equation (2), which is based on this assumption. Second, since the fibers were mounted on the support without preloading, slack often occurred, leading to an increased initial length, $L_{0},$ that was accounted for.

To address this, the real starting point of the load versus displacement curve was determined through the extrapolation for both hemp (Figure 6a) and flax (Figure 6b). The value at which the actual curve began was then added to the theoretical $L_{0}$.

Data processing was conducted using an automated MatLab script. This script was used to compute the conventional stress, *σ*, and strain, *ε*, construct the stress–strain curve, and extract key parameters. The ultimate stress was identified as the maximum value of stress, while the Young’s modulus was determined as the chord modulus of the stress–strain curve within the 0.1–0.2% strain interval, as exemplified in Figure 6c for hemp and Figure 6d for flax.

ASTM C1557-03 [50] provides guidelines for compensating for test frame compliance, but these are intended for linear stress–strain curves. However, for the studied fibers, the stress–strain behavior has been shown to be non-linear, and the significant variation in cross-section would further increase uncertainty if this method would be applied. Furthermore, it has been shown by Engelbrecht-Wiggans that this compensation can be neglected for low-modulus fibers, such as cellulose-based ones [51]. For these reasons, no compliance compensation was introduced in the present study.

### 2.2. Statistical Analysis

Data processing was carried out using various statistical methods. Initially, the mean, standard deviation, and coefficient of variation were calculated for each batch based on the experimental data. Given the high variability in the results, a more in-depth analysis was necessary, which was performed using a two-parameter Weibull distribution. This approach proved to be highly effective for these types of data, offering a better fit than other methods, such as Gaussian modeling [38]. In this study, Weibull modeling was applied to describe the distribution of both tensile strength and Young’s modulus in the tested sample population using the cumulative density function (CDF), as expressed in Equation (3):(3)$$P_{\left(x\right)}=1-e^{{\left(-\frac{x}{\lambda }\right)}^{k}}$$
where the variable x represents the mechanical property (strength or stiffness in this case), $P_{\left(x\right)}$ indicates the probability that a value in the dataset is less than *x*, λ serves as the scale parameter, reflecting the distribution of data across the range of results, and k represents the shape parameter or Weibull modulus.

The value of λ gives an interpretation of how the population is spread across the range of results, with higher values indicating a broader spread, while the value of *k* indicates the function’s width or steepness, with a higher value representing a steeper variation.

Lastly, the quality of the Weibull fit has been calculated by means of the coefficient of determination, R^2^.

## 3. Results and Discussion

### 3.1. Fiber Diameter

The distribution of the minimum fiber diameter along with the histogram gauss fitting is presented in Figure 7. Given that elementary fibers are typically reported to have diameters in the range of 10–25 µm [30,54], the measured values indicate that the tested fibers were primarily technical fibers composed of multiple elementary fibers. In such fibers, the adhesion between individual elementary fibers significantly influences the mechanical properties, often resulting in lower values compared to those of elementary fibers [36].

### 3.2. Mechanical Properties

The stress–strain curves have been constructed using the calculated conventional stress and strain data. Several types of curves have been observed, falling in one of three categories: linear, bi-linear, and tri-linear (Figure 8a). Additionally, in several cases, intermediate fiber failures have been recorded, as shown in the representative curve in Figure 8b.

Failures occurred either abruptly in the fiber’s cross-section or through the rupture of individual elementary fibers; see Figure 9. The latter was the most common cause of failures, reinforcing the classification of the tested fibers as technical fibers.

The average calculated mechanical properties for the tested fibers are presented in Table 4. For hemp, the strength ranges from 412 to 566 MPa, while flax tends to have a higher strength, ranging from 502.04 to 595.65 MPa. Stiffness values follow a similar trend, with those of hemp varying from 22.6 GPa to 35 GPa and those of flax from 31.7 GPa to 50.6 GPa. Both fibers presented high dispersions, from 38.87% for the strength of flax (15 mm) to 76.97% for the strength of hemp (10 mm), generated especially by the rare occurrences of significantly high mechanical properties, which were chosen to be kept in the analysis.

As previously mentioned, this order of magnitude for the dispersion was to be expected, especially since the fibers are extracted from different plants, which have been blended into the initial ballots, and the fiber extraction process influences the properties. It would be an interesting study to analyze the dispersion of the fibers produced by a single plant, by directly extracted it, to remove the aforementioned factors of influence.

### 3.3. Influence of Cross-Section

The significant variability in mechanical properties necessitated a further analysis to identify influencing factors. Therefore, the relationships between stress, stiffness, fiber diameter, and fiber length were examined.

With one exception, all values generally decrease as fiber diameter increases; see Figure 10. At the smallest diameters, the strength and stiffness values closely align with those reported in the literature for elementary fibers [47]. The exception involves 25 mm flax fibers, which show a slight increase in the ultimate stress with an increase in the fiber diameter. However, this increase is minimal, with the linear regression slope at approximately 0.63 MPa/μm, leading to about a 5.2% rise from the regression point at the smallest diameter to the largest measured diameter. The high dispersion and low values of the coefficient of determination, R^2^, show that the variation in mechanical properties cannot be solely linked to the fiber diameter.

The observed decrease in strength with an increase in diameter can be attributed to the statistical likelihood of more defects that could initiate failure. However, this reasoning does not apply to stiffness, as it is determined from the initial portion of the stress–strain curve, i.e., before significant failures occur. A more comprehensive explanation for both phenomena is the increase in interfacial interactions between fibers as the diameter increases. These interfaces have lower mechanical properties than the fibers themselves [36].

### 3.4. Influence of Length

In terms of variation with respect to length, both strength and stiffness are analyzed in Figure 11. For hemp, there is a variation observed between the batches, but no clear trend is observed across the tested lengths. However, for flax, only a slight decrease in strength and stiffness with an increase in length is noted.

When the results of this analysis are combined with the findings from Section 3.2, it appears that, for the tested fiber lengths, any interruption in elementary fibers along their lengths did not significantly impact the results. This suggests that the elementary fibers within the bundles were largely anchored at both ends by the paper support, resulting in minimal variations in properties between batches. This observation partially aligns with Amroune’s findings, which revealed a pronounced and sharp decline in mechanical properties for fiber bundles once their lengths exceeded the average length of the elementary fibers [23].

### 3.5. Weibull Modeling

A representative Weibull CDF for hemp and flax strength is shown in Figure 12. The model fitting reached a coefficient of determination of 0.94 for hemp and 0.97 for flax, indicating a high level of accuracy in the modeling for both cases of hemp and flax strength.

Both hemp and flax strength distributions are effectively modeled by the two-parameter Weibull distribution, with high coefficients of determination ranging from 0.92 to 0.95 for hemp and from 0.95 to 0.97 for flax, Figure 13. Although flax exhibits a slightly higher R^2^ value, the difference is minor and may be due to the inherent variability in the sample populations. Additionally, the modeled curves did not reveal any clear trend in strength distribution with regard to the fiber length, supporting the earlier observation that it did not have a significant impact on the fiber strength within the tested range. The Weibull parameters for both fibers are summarized in Table 5.

In the case of stiffness, as observed in Figure 14, a similar absence of a trend related to fiber length is observed. However, a slight decrease in R^2^ is noted for both fiber types across different lengths, with values ranging from 0.87 to 0.96 for hemp and from 0.92 to 0.97 for flax. Although this decrease is not substantial, it may be attributed to the variation in the cross-sectional area along the length of the fibers, which necessitates an additional simplification for calculating the strain. The parameters for the Weibull modeling of stiffness are summarized in Table 5.

Regarding the evolution for all the modeled curves, skewness towards the left can be observed, suggesting that, while extremely high values are observed for the mechanical properties, these extremely high values occur less often than lower values. Similar curve shapes have been recorded in the literature [33,41], reinforcing the findings of this study. This asymmetry suggests that the Gaussian distribution was indeed not suitable for this type of analysis, as it assumes a symmetric distribution of values around the mean, with no skewness present in the population.

When examining the mechanical properties of the studied fibers alongside those reported in the literature, it becomes evident that fibers cultivated in Romania possess significant potential for producing high-quality composite materials. The consistency of their mechanical performance, coupled with their alignment with documented results, suggests that these locally sourced fibers could play a role in the development of durable and reliable composites. This potential not only highlights the viability of Romanian fibers in advanced material applications but also underscores the opportunity to leverage local resources in the pursuit of sustainable and innovative composite solutions.

## 4. Conclusions and Further Research

This study focused on evaluating the mechanical properties of hemp and flax fibers cultivated in Romania through tensile testing. The aim was to assess their potential as reinforcement materials in composite structures. Statistical analyses were employed to process the experimental data, which included testing samples with four different gauge lengths to investigate the influence of fiber length on mechanical behavior.

Fiber diameter measurements revealed substantial variability, ranging from 20 to 120 µm, with significant fluctuations along the length of the fibers. Tensile testing results indicated two primary modes of failure: abrupt failure across the fiber’s cross-section and individual elementary fiber failures, some of which occurred progressively during testing. The observed stress–strain curves were classified into three categories—linear, bi-linear, and tri-linear—reflecting the typical behavior of vegetable fibers.

The study identified fiber diameter as a significant factor influencing the mechanical properties, while fiber length had a negligible effect. This minimal impact is likely due to the short gauge lengths used in testing, which are comparable to the length of individual elementary fibers. Despite not being specifically cultivated for high strength, both hemp and flax fibers yielded results consistent with documented values for fiber bundles in the literature. Hemp fibers demonstrated an average tensile strength of 475.86 ± 75.49 MPa, while flax fibers showed an average tensile strength of 565.12 ± 44 MPa.

These findings underscore the potential of locally sourced hemp and flax fibers for use in composite materials, particularly for structural applications. The results suggest that further research is warranted to refine these fibers’ integration into composites. Such efforts could lead to the production of high-quality, environmentally friendly composite materials that leverage the strengths of these natural fibers, contributing to more sustainable construction and manufacturing practices.

## Figures and Tables

**Figure 1 materials-17-04871-f001:**
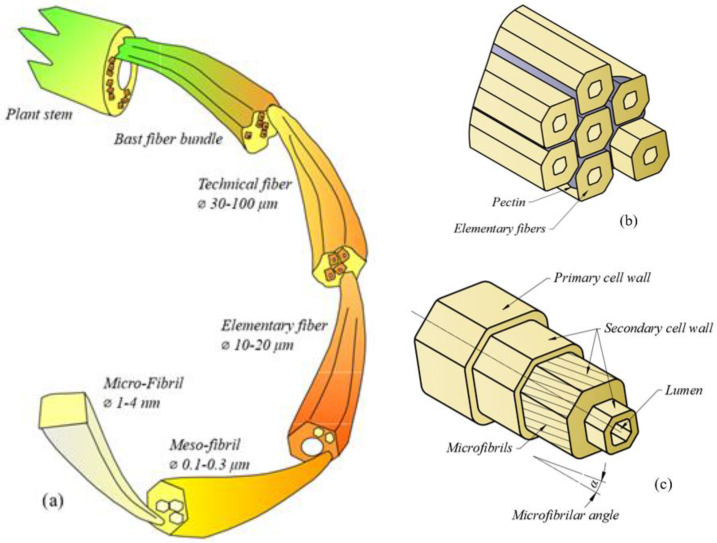
(**a**) Bast plant structure (adapted from [35]); (**b**) structure of fiber bundle; and (**c**) structure of elementary bast fiber.

**Figure 2 materials-17-04871-f002:**
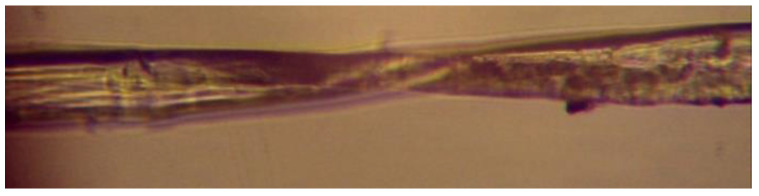
Twist in a technical fiber.

**Figure 3 materials-17-04871-f003:**
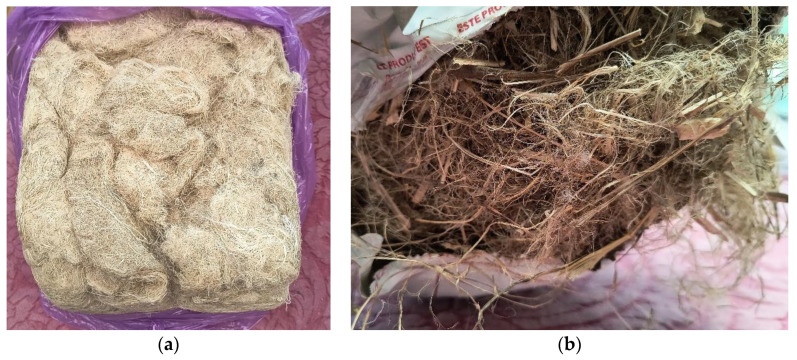
Fiber batch of (**a**) hemp and (**b**) flax.

**Figure 4 materials-17-04871-f004:**
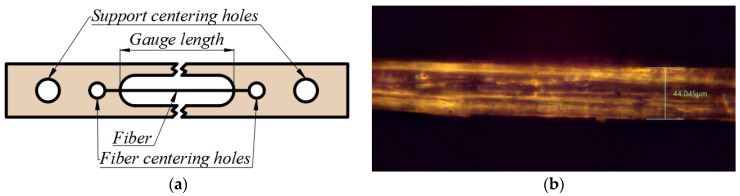
(**a**) Paper support and (**b**) fiber cross-section measurement.

**Figure 5 materials-17-04871-f005:**
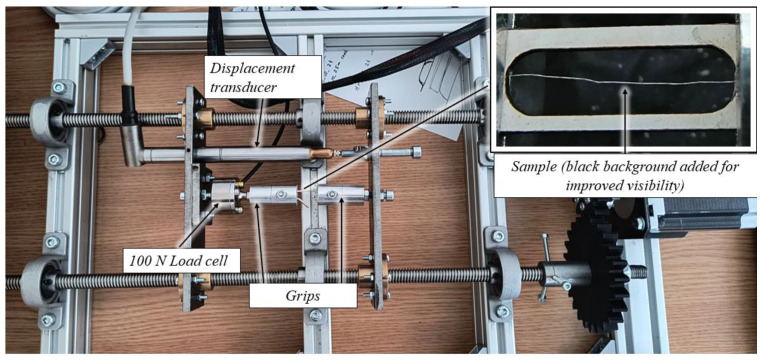
Tensile testing equipment.

**Figure 6 materials-17-04871-f006:**
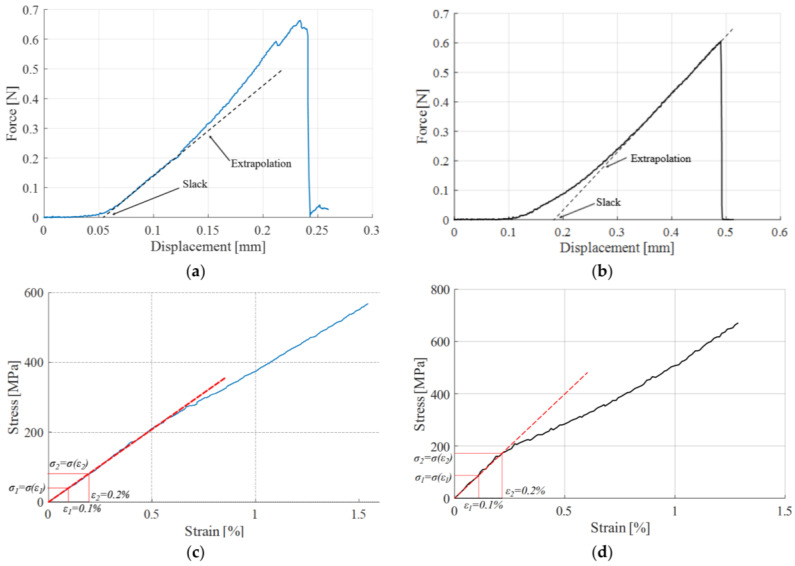
(**a**) Slack extrapolation for hemp; (**b**) slack extrapolation for flax; (**c**) Young’s modulus calculation for hemp; and (**d**) Young’s modulus calculation for flax.

**Figure 7 materials-17-04871-f007:**
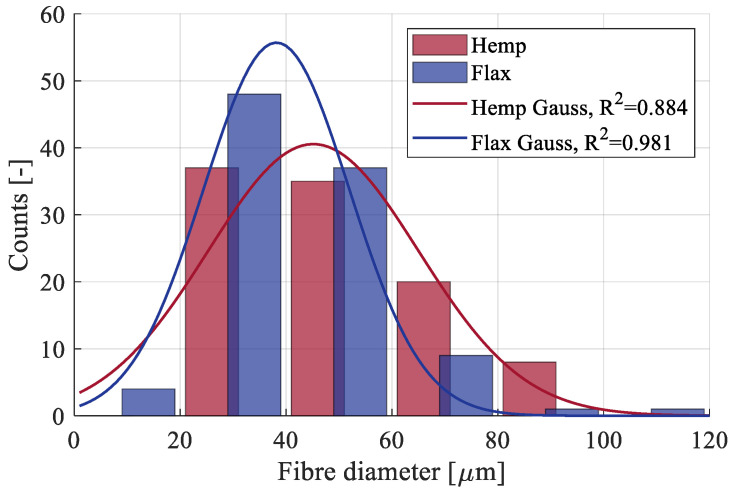
Distribution of the minimum fiber diameter.

**Figure 8 materials-17-04871-f008:**
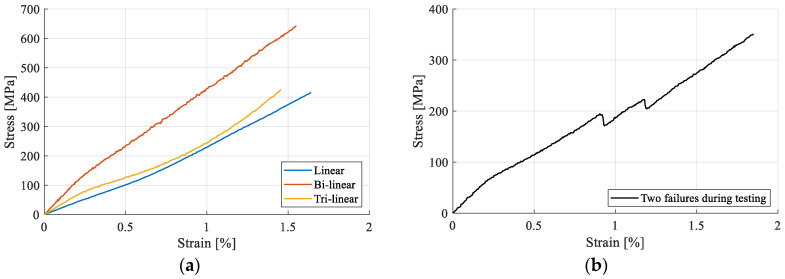
Stress–strain curves: (**a**) types of stress–strain curves (examples from hemp results) and (**b**) fiber failures during tensile testing.

**Figure 9 materials-17-04871-f009:**
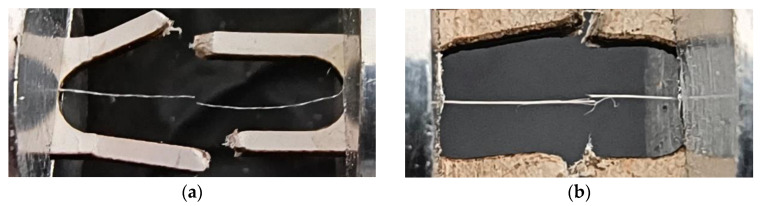
Types of failures: (**a**) in the fiber cross-section and (**b**) through individual elementary fiber failures.

**Figure 10 materials-17-04871-f010:**
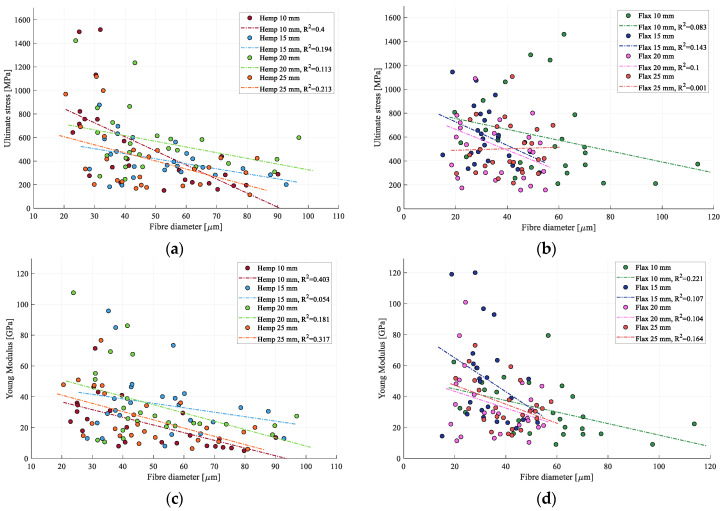
(**a**) Hemp strength with respect to fiber diameter; (**b**) flax strength with respect to fiber diameter; (**c**) hemp stiffness with respect to fiber diameter; and (**d**) flax stiffness with respect to fiber diameter.

**Figure 11 materials-17-04871-f011:**
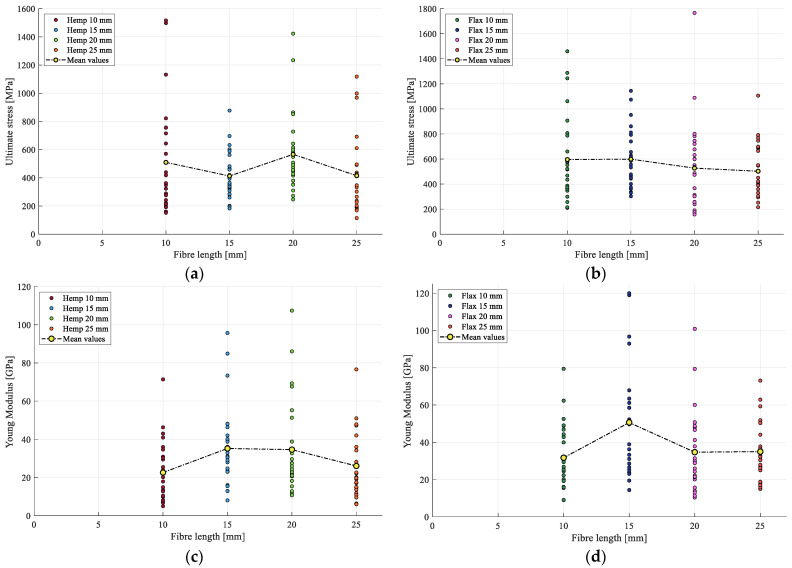
(**a**) Hemp strength with respect to length; (**b**) flax strength with respect to length; (**c**) hemp stiffness with respect to length; and (**d**) flax stiffness with respect to length.

**Figure 12 materials-17-04871-f012:**
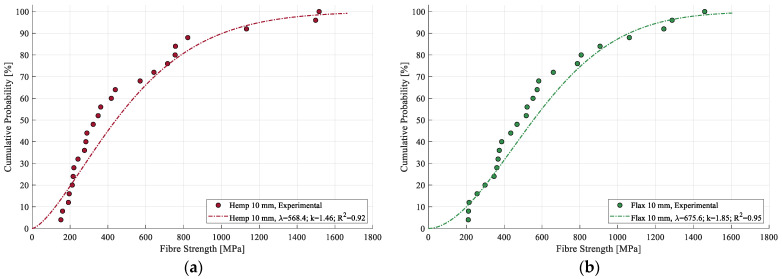
Weibull modeling of experimental data for (**a**) strength of hemp (10 mm) and (**b**) strength of flax (10 mm).

**Figure 13 materials-17-04871-f013:**
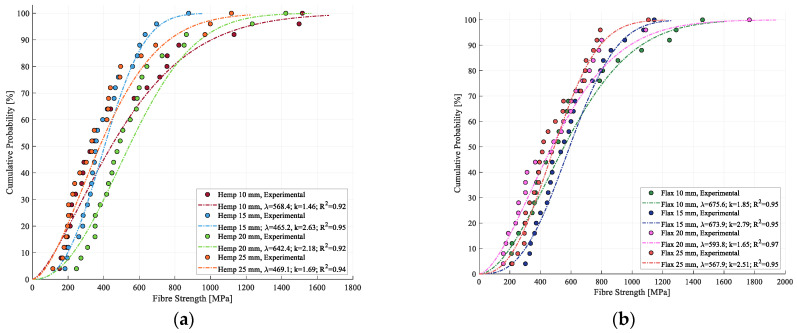
Weibull modeling of cumulative probability for strength of (**a**) hemp fibers and (**b**) flax fibers.

**Figure 14 materials-17-04871-f014:**
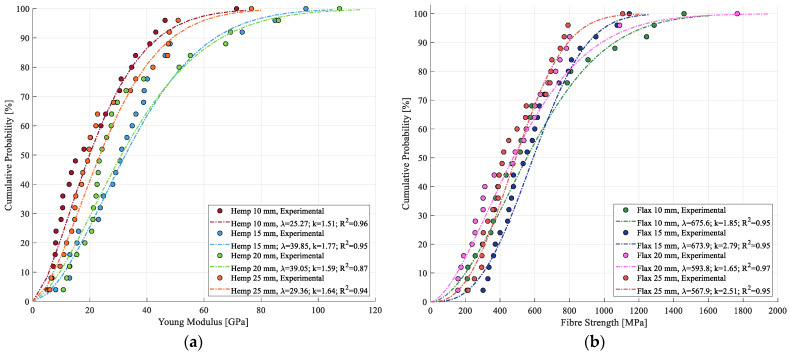
Cumulative probability of Weibull modeling for stiffness of (**a**) hemp fibers and (**b**) flax fibers.

**Table 1 materials-17-04871-t001:** Mechanical properties of hemp fibers.

Diameter	Gauge Length	Strength	Stiffness	Country of Origin	Additional Information	Source
[μm]	[mm]	[MPa]	[GPa]			
-	20	239 ± 93	-	Germany	Short fiber	[42]
67 ± 26	11	277 ± 191	9.5 ± 5.7	The United Kingdom	Short fiber Heat treated	[43]
150 ± 50	10	165 ± 61	6.6 ± 3.22	Romania	Extracted from rope strands	[41]
26.5 ± 6.7	10	514 ± 274	24.8 ± 16.3	The United Kingdom	Retted	[44]
54.3 ± 33	25	700 ± 565	-	The United Kingdom	Cleaned in water prior to testing	[45]

**Table 2 materials-17-04871-t002:** Mechanical properties of flax fibers.

Diameter	Gauge Length	Strength	Stiffness	Country of Origin	Additional Information	Source
[μm]	[mm]	[MPa]	[GPa]			
-	25	540 ± 190	-	The Netherlands	Retted and mechanically extracted	[35]
18.6 ± 3.9	10	1066 ± 342	48.9 ± 12	France	Hermes variety Retted and mechanically extracted	[46]
16 ± 2.7	10	789 ± 276	45.2 ± 12.9	France	Marylin variety Retted and mechanically extracted	[47]
61–122	30	475 ± 170	18.2 ± 6.7	France	Aramis variety Scutched	[48]
220–900	20	695 ± 120	43.2 ± 4.8	France	Aretha variety	[49]

**Table 3 materials-17-04871-t003:** Configuration of tested samples.

	Hemp				Flax			
Gauge length [mm]	10	15	20	25	10	15	20	25
Number of samples	25	25	25	25	25	25	25	25

**Table 4 materials-17-04871-t004:** Mechanical properties of the tested samples.

	Average Strength	CV	Average Stiffness	CV
	[MPa]	[%]	[GPa]	[%]
Hemp (10 mm)	509.46	76.97	22.63	72.02
Hemp (15 mm)	412.56	41.06	35.25	62.05
Hemp (20 mm)	566.45	49.67	34.65	71.19
Hemp (25 mm)	414.96	65.75	26.08	66.95
Flax (10 mm)	595.65	59.42	31.75	56.12
Flax (15 mm)	598.73	38.87	50.67	58.89
Flax (20 mm)	527.04	67.5	34.74	62.33
Flax (25 mm)	502.40	43.3	35.05	44.11

**Table 5 materials-17-04871-t005:** Weibull modeling parameters.

Batch	Ultimate Stress		Young’s Modulus	
	λ	k	λ	k
Hemp (10 mm)	568.4	1.46	25.27	1.51
Hemp (15 mm)	465.2	2.63	39.85	1.77
Hemp (20 mm)	642.4	2.18	39.05	1.59
Hemp (25 mm)	469.1	1.69	29.36	1.64
Flax (10 mm)	675.6	1.85	35.99	1.94
Flax (15 mm)	673.9	2.79	57.46	1.86
Flax (20 mm)	593.8	1.64	39.32	1.77
Flax (25 mm)	567.9	2.51	39.65	2.48

## Data Availability

The original contributions presented in the study are included in the article, and further inquiries can be directed to the corresponding author.

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
