# Peer review of "Mechanical Characterization of Flax and Hemp Fibers Cultivated in Romania"

_materials, 2024, doi:10.3390/ma17194871_

Round 1

Reviewer 1 Report

Comments and Suggestions for Authors

The manuscript with ID: materials-3207901 titled “Mechanical Characterization of Flax and Hemp Fibers Cultivated in Romania” by Stochioiu, C.; et al. is a scientific work where the authors assessed the mechanical properties (ultimate stress and Young’s modulus) of flax and hemp fibers. This knowledge is relevant to design the next-generation of smart composite materials with enhanced performance. The topic is interesting and can reach to audience from all the levels. However, it exists some points that need to be addressed (please, see them below detailed point-by-point) to improve the scientific quality of the submitted manuscript paper before this article will be consider for its publication in Materials.

1) The authors should consider to add the term “Young’s modulus” in the keyword list.

2) “In the effort to minimize environmental impact, the composite materials industry (…) these sustainable sources” (lines 22-28). Here, it may be desirable if the authors could provide quantitative data insights about the worldwide global burdens according to the economic impact of composite materials Industrial sector. This will significantly aid the potential readers to better understand the significance of this research.

3) “Apart the mechanical properties, flax and hemp fibers have several other advantages such as: They are derived from renewable resources (…) involved in production” (lines 70-79). Here, even if I agree with all these statements furnished by the authors, it should not be neglected the excellent interfacial forces exerted between flax and hemp fibers with the composite matrix [1,2] which strengthen the durability of the biobased material.

[1] Marcuello, C.; Chabbert, B.; Berzin, F.; Bercu, N.B.; Molinari, M.; Aguié-Béghin, V. Influence of Surface Chemistry of Fiber and Lignocellulosic Materials on Adhesion Properties with Polybutylene Succinate at Nanoscale. Materials 2023, 16, 2440. https://doi.org/10.3390/ma16062440.

[2] Huang, S.; Fu, Q.; Yan, L.; Kasal, B. Characterization of interfacial properties between fibre and polymer matrix in composite materials – A critical review. J. Mater. Res. Technol. 2021, 13, 1441-1484. https://doi.org/10.1016/j.jmrt.2021.05.076.

4) Figure 2 (line 134). It should be also mentioned the employed technique to gather this Image.

5) Table 1 (line 149). The significant figures need to be homogenized. Similar comment for the Table 3 (line 267) where the data units displayed in the Average Stiffness column should be exchanged from MPa to GPa in line with other part sections of this manuscript.

6) Figure 6 (line 218). Representative force-displacement and stress-strain curves should be added related to those measurements carried out in flax and hemp fibers. Results related to hemp fibers were already stated in the Fig. 8 (line 256).

7) Figure 7 (line 245). The standard deviation bars need to be added for each examined fibre diameter condition. Then the Gaussian fitting needs also to be displayed for both distributions.

8) Figure 10 (line 290). The regression coefficient (R2) needs to be displayed for each linear distribution.

9) Conclusion and further work (lines 351-377). This section perfectly remarks the most relevant outcomes found by the authors in this work and the potential future action lines to pursue this research. No actions are requested from the authors.

Comments on the Quality of English Language

The manuscript is generally well-written albeit it would be advisable if the authors could do a final check in order to polish those final details susceptible to be improved before its resubmission process.

Reviewer 2 Report

Comments and Suggestions for Authors

Dear Authors, 

In my opinion the article seems interesting but needs corrections. 

- The introduction needs to be rewritten for the title. The introduction contains a lot of commonly known information. In the title you write about hemp and flax grown in Romania, focus on that.

The topic of the manuscript concerns mechanical characterization of flax and hemp fibers cultivated in Romania, but unfortunately we don't know anything about these fibers. Please provide the plant varieties from which the fibers come. Please provide the most important information, namely what method was used to obtain the fibers?. Are these dew retted or water retted fibers maybe different? What kind of fibers are they, long or short? Why do they differ in color and length?

Page 2, lines 74-75.  “The technology for extracting the fibers is well-established, with only minor adjustments required to cater to industry-specific needs” – please explain what exactly do you mean.

Page 2, line 91-92.   “Unlike elementary fibers, which are often studied in isolation..”, Please provide the references.

 Page 4, lines . “Studies on natural fibers are presented in literature, covering various varieties, regions, and methods of cultivation or extraction”. - You did not include such information in your work. In my opinion they should be included.

Page 5, lines 171-173. “Before sample preparation, the fibers were cleansed using a soap and water solution  to minimize impurities and the amount of potentially degrading bacteria, following a frequent practice in the retting process”. Does the standard ASTM C1557 include information about preparing the fiber in the way you describe? Does the tests were conducted on wet or dried samples?

Page 9, line 9, hemp, Flax - please standardize the spelling.

Round 2

Reviewer 1 Report

Comments and Suggestions for Authors

The authors did a great deal of effort to cover the suggestions raised by the Reviewers. For it, the scientific manuscript quality was greatly improved.

Al latest remark and linked with the previous point 3, the authors answered:  "They possess better fiber/matrix bonding than glass fibers [26]”.

It needs to be also referred the strong fiber/matrix bonding [1]:

[1] Marcuello, C.; Chabbert, B.; Berzin, F.; Bercu, N.B.; Molinari, M.; Aguié-Béghin, V. Influence of Surface Chemistry of Fiber and Lignocellulosic Materials on Adhesion Properties with Polybutylene Succinate at Nanoscale. Materials 2023, 16, 2440. https://doi.org/10.3390/ma16062440

Author Response

The authors did a great deal of effort to cover the suggestions raised by the Reviewers. For it, the scientific manuscript quality was greatly improved.

Al latest remark and linked with the previous point 3, the authors answered:  "They possess better fiber/matrix bonding than glass fibers [26]”.

It needs to be also referred the strong fiber/matrix bonding [1]:

[1] Marcuello, C.; Chabbert, B.; Berzin, F.; Bercu, N.B.; Molinari, M.; Aguié-Béghin, V. Influence of Surface Chemistry of Fiber and Lignocellulosic Materials on Adhesion Properties with Polybutylene Succinate at Nanoscale. Materials 202316, 2440. https://doi.org/10.3390/ma16062440

Thank you for the comment. After further reviewing the litterature, we have modified the text by replacing a previous statment, "They possess better fiber/matrix bonding than glass fibers [26];" with "They possess better fiber/matrix bonding properties when compared to other vegetable fibers [26];"